# Effects of PEF on Cell and Transcriptomic of *Escherichia coli*

**DOI:** 10.3390/microorganisms12071380

**Published:** 2024-07-07

**Authors:** Jinyan Kuang, Ying Lin, Li Wang, Zikang Yan, Jinmei Wei, Jin Du, Zongjun Li

**Affiliations:** 1Hunan Province Key Laboratory of Food Science and Biotechnology, Changsha 410128, China; 18890577470@163.com (J.K.); linying2310@163.com (Y.L.); 15285223720@163.com (L.W.); yanzikang11@163.com (Z.Y.); 18178431439@163.com (J.W.); 19507478621@163.com (J.D.); 2College of Food Science and Technology, Hunan Agricultural University, Changsha 410128, China

**Keywords:** pulsed electric field (PEF), *Escherichia coli*, non-thermal sterilization, inactivation mechanism, transcriptome

## Abstract

Pulsed electric field (PEF) is an up-to-date non-thermal processing technology with a wide range of applications in the food industry. The inactivation effect of PEF on *Escherichia coli* was different under different conditions. The *E. coli* inactivated number was 1.13 ± 0.01 lg CFU/mL when PEF was treated for 60 min and treated with 0.24 kV/cm. The treatment times were found to be positively correlated with the inactivation effect of PEF, and the number of *E. coli* was reduced by 3.09 ± 0.01 lg CFU/mL after 100 min of treatment. The inactivation assays showed that *E. coli* was inactivated at electrical intensity (0.24 kV/cm) within 100 min, providing an effective inactivating outcome for Gram-negative bacteria. The purpose of this work was to investigate the cellular level (morphological destruction, intracellular macromolecule damage, intracellular enzyme inactivation) as well as the molecular level via transcriptome analysis. Field Emission Scanning Electron Microscopy (TFESEM) and Transmission Electron Microscope (TEM) results demonstrated that cell permeability was disrupted after PEF treatment. Entocytes, including proteins and DNA, were markedly reduced after PEF treatment. In addition, the activities of Pyruvate Kinase (PK), Succinate Dehydrogenase (SDH), and Adenosine Triphosphatase (ATPase) were inhibited remarkably for PEF-treated samples. Transcriptome sequencing results showed that differentially expressed genes (DEGs) related to the biosynthesis of the cell membrane, DNA replication and repair, energy metabolism, and mobility were significantly affected. In conclusion, membrane damage, energy metabolism disruption, and other pathways are important mechanisms of PEF’s inhibitory effect on *E. coli*.

## 1. Introduction

Foodborne diseases caused by bacterial contamination are among the most important problems affecting human public health and food safety [1]. *Escherichia coli* is a common foodborne pathogen belonging to the Gram-negative group that can be infected through the respiratory and digestive tracts, triggering diarrhea, vomiting, abdominal pain, and rapid dehydration, posing a serious threat to both consumers and the food industry [2]. In recent years, a large number of *E. coli* have been detected successively in various types of foods around the world, mainly in cucumber [3], flour [4], milk [5], beef [6], and other foods. An outbreak of Shiga-toxin-producing *Escherichia coli* O157:H7 occurred in the United Kingdom in August 2020. A total of 36 cases were identified [3]. Sun [7] collected 15 different types of food products (495 samples in total) for retail sale in Beijing, China. It was found that food animal products (66.70%), vegetables (46.20%), aquatic products (22.20%), and fruits (14.40%) were highly contaminated with *E. coli*. Simultaneously, the bacteria can cause widespread diseases (environmental pollution) and readily make its way into the human food chain through soil crops. The potential threats to food safety from microbiological factors cannot be ignored, making sterilization an essential process in the food industry. Sterilization technology not only extends the shelf life of food but also reduces the risk of contamination by pathogenic bacteria [8].

Traditional thermal food sterilization technology has caused significant damage to the functional components of food as well as to the organoleptic quality of food and cannot meet the demands of modern food processing [9]. Thermal sterilization has been used by the food industries to inactivate foodborne pathogens, including bacterial spores, for the extension of the shelf life of the food [10]. Food scientists have always been challenged to eliminate pathogens and spores in food without producing any undesirable changes, and this has received much attention. As a complement or substitute to traditional thermal technology, pulsed electric field (PEF) is a promising non-thermal food processing technology for microorganism inactivation with minimal impact on sensory and nutritional food properties [11,12,13]. PEF has been the most researched non-thermal technology until now. But the current PEF sterilization technology is difficult to industrialize. PEF devices are bulky, expensive, and require a higher voltage, and electrodes are prone to electrolysis and higher risks during operation at high voltage [14]. PEF is considered one of the most promising non-thermal sterilization techniques for inactivating pathogenic and spoilage bacteria in liquid and semi-solid foods, as charged molecules are present in liquids and electric currents move more efficiently in liquid foods [15]. Recently, moderate intensity pulsed electric field (PEF) has attracted the attention of most researchers [16,17,18], which reduces energy consumption, is simple to operate, and is low risk (electric field intensity > 5 kV/cm). However, up to date, most of the research is limited to the study of the physiological and biochemical effects of electric field technology on the cells of living organisms; there is no in-depth investigation of the mechanism of its action, and the study of the mechanism of inactivation of microorganisms by PEF is less at the molecular level. In-depth studies are necessary for a better understanding of the killing of bacteria by PEF treatments.

Transcriptomics is the comprehensive analysis of the entire set of transcripts of a given cell, tissue, organ, or whole organism at a particular time or developmental step, or possibly under certain specific physiological conditions, revealing the molecular mechanisms involved in specific biological processes [19]. Transcriptomics is a discipline that studies the expression of genes and the regulation of transcription. It can study changes in microbial gene expression at the molecular level. Therefore, in recent years, transcriptomics has been widely used to study the mechanism of action of bacteriostatic substances on foodborne pathogens [20]. Currently, transcriptomics has been used to study microbial adaptations and stress responses under diverse stress conditions. Transcriptome profiling of bacterial cells treated with PEF would help to understand the mechanism of inactivation and the resistance mechanisms that cells use to cope with PEF.

*E. coli* might be inhibited by PEF through multiple mechanisms, based on the above information. The *E. coli* response to PEF was investigated to validate this prediction. FESEM and TEM were recruited to observe the morphological changes of *E. coli* exposed to PEF treatment. Leakage of cytoplasmic contents is a classic indication of damage to the bacterial cytoplasmic membrane. Enzyme activities were analyzed to determine the influences of PEF on cellular metabolisms. In particular, transcriptomic analysis was conducted to determine the differentially expressed genes (DEGs) of PEF-treated *E. coli* versus the wild type and to explore the potential modes of action. A combination of the above approaches was used to systematically examine how PEF inhibits *E. coli*. The mechanism of sterilization is expected to perfect the theoretical study of PEF. The mechanism of PEF on *E. coli* is important for the application of non-thermal sterilization technology in the food industry.

## 2. Materials and Methods

### 2.1. Reagents and Bacterial Strains

*Escherichia coli* ATCC 25922 was purchased from the China Microbial Strain Conservation Center (CMSC) (Beijing, China) and cultured on Tryptone Soy Broth (TSB) (Guangzhou, China) at 37 °C. Enzymes (Adenosine Triphosphatase (ATP), Pyruvate Kinase (PK), and Succinic Dehydrogenase (SDH)) were measured according to the assay kits manual from Nanjing Jian Cheng Bioengineering Institute (Nanjing, China). HiScript II Q RT SuperMix for qPCR (+gDNA wiper) and ChamQ universal SYBR qPCR Master Mix were measured according to the assay kits manual from Nanjing Vazyme Biotech Co., Ltd. (Nanjing, China).

### 2.2. Instruments and Equipment

High-voltage pulse electric field equipment (self-made by South China University of Technology (Guangzhou, China)), LMQ.C Automatic Autoclaving Steam Sterilization Cooker (Shandong Xinhua Medical Device Co., Ltd. (Zibo, China)), FiveEasyPlusTM Desktop conductivity meter (Mettler Toledo (Changzhou) Precision Instruments Co., Ltd. (Changzhou, China)), FEI TECNAI G2 12 Biotransmission Electron Microscopy (Beijing, China), SU8020 emission scanning electron microscope (Hitachi, Tokyo, Japan), SpectraMax^®^ ABS Plus Microplate reader (Meigu Molecular Devices (Shanghai) Co., Ltd. (Shanghai, China)), 580BR MyCyclerTM Thermal Cycler (Bio-Rad, Hercules, CA, USA), and R0918102 Roter-Gene Q (QIAGEN, Duesseldorf, Germany) were used.

### 2.3. Methods

#### 2.3.1. Bacterial Suspension Preparation

A small number of bacteria were picked from the test tube slant strains kept in the refrigerator at 4 °C, inoculated into 50 mL of liquid medium under aseptic conditions, and incubated at 37 °C, 120 rpm for 22 h to obtain the microbial culture solution. The cultured microbial culture solution was inoculated into a conical flask of 50 mL of liquid medium in an amount of 1 mL and incubated at 37 °C and 120 rpm for 6 h to the logarithmic growth stage. The cultured solution was centrifuged at 5000 rpm for 10 min to collect the bacterium, and then the concentration of the bacterial solution with a mass fraction of 0.85% NaCl and the bacterial suspension was reserved for use.

#### 2.3.2. Evaluation of the Effect of PEF on the Inactivation of *E. coli*

The sample chamber was filled with a volume of 50 mL of bacterial suspension. Effects of different solution media, NaCl mass fraction, electric field intensity, treatment time, and bacterial density on microbial inactivation were investigated. According to the findings, the following treatment parameters were used for the *E. coli* inactivation mechanism and transcriptome analysis by PEF: 100 min of treatment time, 0.24 kV/cm of electric field intensity, 10^7^ CFU/mL of bacterial density, a mass fraction of 0.85% NaCl solution medium. PEF is an exponential decay waveform with a reference time of 25.00 μs, an exponential decay waveform occupying 15.00 μs, a trigger source of 4.00 V, and a trigger frequency of 10.0 Hz. The conditions in the one-way test were as follows: different media solutions (0.20 kV/cm of electric field intensity, 60 min of treatment time, 10^7^ CFU/mL of bacterial density), different NaCl mass fractions (0.20 kV/cm of electric field intensity, 60 min of treatment time, 10^7^ CFU/mL of bacterial density), different electric field intensities (mass fraction of 0.85% NaCl solution, 60 min of treatment time, 10^7^ CFU/mL of bacterial density), different treatment times (mass fraction of 0.85% NaCl solution, 0.20 kV/cm of electric field intensity, 10^7^ CFU/mL of bacterial density), and different bacterial density (mass fraction of 0.85% NaCl solution, 0.20 kV/cm of electric field intensity, 60 min of treatment time).

A total of 1 mL of the liquid suspension with the PEF group and the control group was absorbed. After gradually diluting aseptic saline and pouring it into a plate, colonies were counted for 48 h in a 37 °C incubator. The results were expressed as lgN_0_-lgN, where N_0_ and N denoted the number of microorganisms (CFU/mL) before and after PEF treatment, respectively. Using an electronic thermometer to detect temperature simultaneously disproves the idea that PEF is a heat-sterilization technique and eliminates the impact of temperature on *E. coli*.

#### 2.3.3. Field Emission Scanning Electron Microscopy and Transmission Electron Microscope

The morphological changes could be observed by FESEM according to the reported method with some modifications [21]. *E. coli* was dehydrated in ethanol (50%, 70%, and 100% (*v*/*v*) for 10 min), washed three times with PBS (0.01 mol/L, pH 7.4), and fixed with 2.50% (*v*/*v*) glutaraldehyde overnight at 4 °C. The samples were then sprayed with gold, vacuum-dried for one hour at −20 °C, and examined. The electron beam’s acceleration voltage was 5 kV.

After fixing *E. coli* for the night at 4 °C with 2.50% (*v*/*v*) glutaraldehyde, the bacteria were washed three times for 15 min with PBS (0.01 mol/L, pH 7.4) for 15 min. After fixing *E. coli* for 1 to 2 h with 1% (*v*/*v*) osmium tetroxide, it was washed three times for 15 min with PBS (0.01 mol/L, pH 7.4). The samples were dehydrated in three different solutions: saturated ethanol (30%, 50%, 70%, 80%, 90%, 95%, and 100% (*v*/*v*) for 15 min), saturated solution (acetone: embedding solution (*v*:*v* = 1:1 for 1 h)) and saturated solution (acetone: embedding solution (*v*:*v* = 3:1 for 3 h)). The samples were submerged in pure embedding liquid overnight. The permeated samples were embedded and heated at 70 °C overnight to obtain the embedded sample. After the material was cut with a microtome, slices between 70 nm and 90 nm were produced. The slices were exposed to lead citrate dye and uranyl acetate dye for five minutes each, after which they were dried and subjected to an accelerated voltage of 100 kV for camera observation.

#### 2.3.4. Leakage of Intracellular Material

The protein and nucleic acid compositions were adapted and adjusted from earlier methods [22]. Briefly, 50 mL of bacterial suspension before and after electric field treatment was centrifuged at 5000 rpm for 10 min at 4 °C. The supernatant was collected, and 3 mL of supernatant was placed in a quartz cuvette. The absorbances at 260 nm and 280 nm were used to quantify the protein and nucleic acid leakage in bacteria exposed to PEF, respectively.

#### 2.3.5. Enzyme Activity Analysis

The bacterial fluids with or without PEF treatment were centrifuged at 5000 rpm for 10 min, twice cleaned with 0.85% NaCl, and the precipitated bacteria were then collected. Following resuspension, the bacterial suspension was crushed in an ice bath by an ultrasonic sonifier (300 W, 2 s of sonication, 9.9 s of sonication interval, total 7 min). The supernatant was collected after centrifugation. The protein concentration was measured using the BCA protein concentration assay kit. The subsequent experimental procedures were performed following kit (PK, SDH, and ATPase) instructions [23].

#### 2.3.6. Transcriptomic Analysis

To evaluate the transcriptional changes of bacteria in response to PEF, *E. coli* was treated with PEF for 100 min. The quality evaluation of the samples from the control group and the PEF treatment group was performed by Shanghai Meiji Biomedical Technology Co., Ltd. (Shanghai, China). All mRNAs transcribed by *E. coli* were sequenced using the Illumina Novaseq 6000 Sequencing Platform (Shanghai, China). The sequencing experiment was conducted using the Illumina TruseqTM RNA Sample Prep Kit method to construct the library. Gene level transcripts per million reads (TPM) were generated using RSEM in order to identify the differentially expressed genes (DEGs) between the samples. TPM is calculated in terms of the number of transcript strips, using the number of transcript strips instead of the number of spliced fragments, taking into account factors such as transcript length and the number of genes expressed in the sample. DEGs were then screened using DESeq2 with |log_2_FC| ≥ 1.0 and *p*-adjust < 0.05. Next, using the Kyoto Encyclopedia of Genes and Genomes (KEGG) and Gene Ontology (GO) for hypergeometric distribution testing, DEGs were further exposed to pathway enrichment analysis and function annotation.

#### 2.3.7. Quantitative Reverse Transcription PCR (RT-qPCR) Analysis

RT-qPCR was performed to verify the changed expression levels of the related DEGs. The RT-qPCR primers were designed and determined by Bioengineering Co., Ltd. (Shanghai, China). Target genes were selected (genes with significant expression differences in the treatment group compared to the control group), and the sequence information of those genes was obtained from RNA-Seq analysis. (Table 1). 16SrRNA was selected as the reference gene, and each sample was run in triplicate. Using the Trizol kit, total RNA from *E. coli* was extracted, and the HiScript ll Q RT SuperMix for qPCR (+gDNA wiper) kit was used to convert it to cDNA. The RT-qPCR was performed using the ChamQ universal SYBR qPCR Master Mix kit and the Rotor-Gene Q equipment. The relative expression of 2^−ΔΔCt^ expressed genes was determined by analyzing the data. The RT-qPCR reaction mixture consisted of 10 μL ChamQ universal SYBR qPCR Master Mix, 0.4 μL of each primer (10 μM), 2 μL of cDNA and 7.2 μL of ddH_2_O. Cycling parameters for RT-qPCR included an initial denaturation at 95 °C for 30 s, followed by 40 cycles at 90 °C for 10 s, 60 °C for 30 s, and then melting curve analysis at 95 °C for 5 s, 60 °C for 90 s and followed by heating from 60 to 95 °C with continuous fluorescence collection. The 2^−ΔΔCT^ method was used to calculate the changes in DEG expression.

#### 2.3.8. Statistical Analysis

All experiments were repeated at least three times to acquire the value of average ± standard deviation (SD). The significance analysis was assessed at the level of *p* < 0.05 by SPSS Statistics 25 software (IBM Co., Chicago, IL, USA). Graphs were drawn with Origin 2021 software (OriginLab Inc., Northampton, MA, USA).

## 3. Results and Discussion

From the above Table 2, it can be seen that the size of the conductivity values of different media is different. This table provided a reference for the effectiveness of subsequent PEF in the inactivation of *E. coli*. PEF treatment was less effective in inhibiting microorganisms in high conductivity solutions.

### 3.1. Inactivation of E. coli by Pulsed Electric Field

#### 3.1.1. Inactivation of *E. coli* in Different Media Solutions by Pulsed Electric Field

In this study, NaCl solution, PBS, and peptone solution were used as treatment media to study the effect of different treatment media on the killing of *E. coli* by PEF (0.20 kV/cm of electric field intensity, 60 min of treatment time, 10^7^ CFU/mL of bacterial density). The bactericidal effect of PEF on *E. coli* was different under different media conditions. As shown in Figure 1A, at the same electric field strength and treatment time, *E. coli* presented a higher level of inactivation in the mass fraction of NaCl, and the number of microorganisms decreased by 0.96 ± 0.87% lg CFU/mL. This indicates that lower conductivity favors the bactericidal efficiency of PEF. Among other things, PBS has a salt balance, an adjustable pH buffering effect, and a salt ion concentration that maintains the steady state of the cell membranes of organisms. The ionic strength of the peptone solution is small, and the conductivity of the solution is low. Conductivity is one of the most vital factors affecting the sterilizing effect of the electric field, which determines the resistance of the processing chamber and thus affects the field strength and temperature [24]. Because ionic strength affects cell membrane stability based on variations in cytoplasmic ionic concentration and external media conductivity, it can also impact inactivation outcomes [25].

#### 3.1.2. Inactivation of *E. coli* in Different NaCl Mass Fractions by Pulsed Electric Field

To evaluate the effect of conductivity on the inactivation of PEF treatment, solutions with different mass fractions of NaCl were used (0.20 kV/cm of electric field intensity, 60 min of treatment time, 10^7^ CFU/mL of bacterial density). *E. coli* was more tolerant to salt and grew normally in a mass fraction of 3.00% NaCl solution compared to low salt solution (Figure 1B). The results showed that the bactericidal effect of *E. coli* was slightly lower in the solution with a high mass fraction of NaCl. The conductivity of the mass fraction of 3.00% NaCl solution was higher compared to the mass fraction of 0.85% NaCl solution, with conductivities of 15.06 ± 0.01 and 47.21 ± 0.02 mS/cm, respectively. Conductivity determines the resistance of the load at the discharge end, thus affecting the field strength and temperature, and low conductivity has a strong bactericidal effect [26]. With the increase in the solution mass fraction, the numbers of inactivation were 0.97 ± 0.38%, 0.76 ± 0.68%, 0.24 ± 0.01, and 0.17 ± 0.93% lg CFU/mL, and the temperature gradually increased with temperature differences of 1.20, 1.40, 1.80, and 2.00 °C, respectively. PEF showed suitable inactivation of *E. coli* in a mass fraction of 0.85% NaCl solution. PEF treatment is the most effective for the microbial inhibition of foods with low electrical conductivity.

#### 3.1.3. Inactivation of *E. coli* in Different Electric Field Intensities by Pulsed Electric Field

As can be seen from the diagram 1C (mass fraction of 0.85% NaCl solution, 60 min of treatment time, 10^7^ CFU/mL of bacterial density), the lethality number of *E. coli* increased significantly with the increase in electric field intensities (0.16–0.24 kV/cm), indicating that the change in electric field intensity had a significant impact on the number of *E. coli* lethality in sodium chloride solution. *E. coli* inactivated numbers were 0.14 ± 0.01, 0.95 ± 0.01, and 1.13 ± 0.01 lg CFU/mL when PEF was treated for 60 min and treated with 0.16, 0.20, and 0.24 kV/cm, respectively, with temperature differences of 0.90, 1.20, and 3.50 °C between treatment and pre-treatment. The mechanism of electroporation and its effects on the inactivation of microorganisms has been explained by the microbial cell membrane dielectric breakdown theory. When an electric field strength is applied above a threshold, irreversible cell membrane breakdown occurs, resulting in cell death [27]. When microbial cells are continuously exposed to an electric field, a large amount of electrical charge accumulates on both sides of the cell membrane. The difference in electrical potential between the two sides of the cell membrane changes, and cell membrane permeability increases, leading to cell death [28]. Caminiti [29] reported that apple juice was processed using the lab-scale PEF system equipment and employing mono-polar pulses with a pulse width of 1 μs. When PEF was applied individually, the number of reductions observed was significantly decreased for PEF.

#### 3.1.4. Inactivation of *E. coli* in Different Treatment Times by Pulsed Electric Field

The bactericidal effect increased with the prolongation of the electric field treatment time (20–100 min) (Figure 1D). The effect of PEF inactivation of *E. coli* was significantly enhanced (mass fraction of 0.85% NaCl solution, 0.20 kV/cm of electric field intensity, 10^7^ CFU/mL of bacterial density). Under a certain electric field strength (0.20 kV/cm), the best inactivation effect of PEF on *E. coli* was achieved at a treatment time of 100 min, with a reduction of 3.09 ± 0.01 lg CFU/mL and a temperature difference of 2.10 °C. The number of *E. coli* was only reduced by 0.16 ± 0.02 lg CFU/mL during the 20 min treatment, and the temperature difference was 0.40 °C. With the passage of time, microbial cells are continuously exposed to the electric field, the potential difference between the two ends of the cell membrane will change accordingly, and the permeability of the cell membrane will change, which will lead to the death of the cells. It can be seen that electric field intensity and pulse treatment time are two important factors that affect the inactivation effect of PEF on *E. coli*, which is consistent with the results reported by Gabrić [30,31].

#### 3.1.5. Inactivation of *E. coli* in Different Bacterial Density by Pulsed Electric Field

Under the same conditions of PEF strength and treatment time, the effect of bacterial density on *E. coli* is shown in Figure 1E (mass fraction of 0.85% NaCl solution, 0.20 kV/cm of electric field intensity, 60 min of treatment time). When the density of bacteria was 10^5^ CFU/mL, the bactericidal effect was the best; the number decreased by 1.53 ± 0.05 lg CFU/mL. When the bacterial density was 10^9^ CFU/mL, only 0.18 ± 0.61% lg CFU/mL was decreased compared with the control group. The results showed that the inactivating efficacy was enhanced by a decrease in the density of the bacteria solution under the fixed condition. At low cell densities, single cells are dispersed, and bacteria are arranged in a monolayer; at higher cell densities, bacteria are presented in a multilayer, small cluster, island, or web structure. Under PEF treatment, the top layer of bacteria, even if inactivated, can act as a physical barrier, protecting the underlying bacteria from electric fields and thereby reducing microbial survival [32]. Fernández [33] studied the effect of microbial loading on the efficiency of cold atmospheric gas plasma inactivation of *Salmonella enterica* serovar Typhimurium. The results investigated that the inactivation rate of *S. Typhimurium* was inversely proportional to the initial bacterial concentration.

#### 3.1.6. Inactivation of *E. coli* by Pulsed Electric Field Treatment

The number of *E. coli* decreased significantly (*p* < 0.05) after PEF treatment (Figure 1F) (100 min of treatment time, 0.24 kV/cm of electric field intensity, 10^7^ CFU/mL of bacterial density, a mass fraction of 0.85% NaCl solution medium). The number of *E. coli* in the control group was 7.27 ± 0.01 lg CFU/mL, and the number of *E. coli* plummeted to 3.62 ± 0.03 lg CFU/mL after PEF. The media solution’s temperature increased by 6.30 °C after the PEF treatment. The study by Wang [34] suggested that yeast cells may be harmed by low doses of electric fields. The reduction in *E. coli* by electric field below 50 °C was caused by its non-thermal effect [35].

### 3.2. Morphological Changes of E. coli

The morphological and physical changes of *E. coli* were treated with PEF for 100 min. The microstructure of *E. coli* was visualized, and the results are presented in Figure 2. In the control group, all of the *E. coli* cells had a complete rodlike border (Figure 2A,B); however, some of the bacteria that were subjected to PEF processing had wrinkles, irregularities, and damaged parts of the cell. (Figure 2C,D).

The TEM pictures showed a similar pattern to FESEM results. As seen in Figure 3, the untreated bacteria had a smooth, integrated cell membrane, integrated cell wall, and slippy surface. However, after being treated with PEF for 100 min, the cell structure of *E. coli* underwent significant changes. Some of the cells had fractured cell walls, the cytoplasm inside the cells was not homogeneous and appeared as a cavity, the cell wall’s structure was fuzzy, the cell wall appeared discontinuous, and some of the cells had completely disintegrated and fragmented.

The FESEM and TEM results demonstrated that PEF treatment played a destructive effect on the cytomembrane and altered the cell structure of *E. coli*, which can be regarded as the antimicrobial mechanism of PEF against *E. coli* [16,23]. This indicated that the PEF might have detrimental effects on the cytoplasmic membrane and cell wall.

### 3.3. Effect of Pulsed Electric Field on Bacterial Cell Membrane Permeability

The OD 260 and OD 280 values of the supernatant of the treatment solution are shown in Figure 4A. The absorbances at 260 and 280 nm were used to estimate the protein and DNA concentrations [36]. The OD 260 and OD 280 values significantly increased (*p* < 0.05) by PEF treatment as compared with the control group. The OD 260 and OD 280 values were 0.35 ± 0.05% and 0.31 ± 0.00 after PEF treatment compared with 0.10 ± 0.00 and 0.09 ± 0.00 in the control group. The results were consistent with the study of Zhu [37] and Coustets [38] on PEF treatment.

The nucleic acids and proteins in the bacterial supernatant were significantly enhanced with the increase in PEF treatment time. Leakage of cytoplasmic contents is a classic indication of damage to the bacterial cytoplasmic membrane. The bacterial membrane serves as a structural component. Small ions like potassium and phosphate have a tendency to leak out of damaged cell membranes first, then big molecules like DNA, RNA, protein, and other substances. Nucleic acids have characteristic peaks in the ultraviolet region at 260 nm because of their purine and pyrimidine rings, while proteins have characteristic absorption peaks at 280 nm because of the tryptophan and tyrosine residues they contain. The concentration of each material is directly correlated with the absorbance value [22]. According to recent research, PEF of 7 kV/cm with 10 μs long pulses might result in irreversible membrane holes on cells with a minimum radius of 5.10 nm. This finding provided a potential explanation for the observed increase in membrane permeability [39]. Our experiments demonstrated that PEF treatment caused significant and irreversible damage to the bacterial structure and cell membrane permeability, thus resulting in the leakage of nucleic acids and proteins from cells.

### 3.4. Effect of Pulsed Electric Field on Pyruvate Kinase, Succinate Dehydrogenase, and Adenosine Triphosphatase

As demonstrated in Figure 4B–D, PK, SDH, and ATPase activities were all decreased by PEF treatment. PK activity decreased after PEF treatment, and the activity after 100 min of treatment was 18.46 ± 4.57 U/g protein. Furthermore, SDH activity was found to be 6.03 ± 0.00 U/mg protein in treated cells and 65.26 ± 2.51 U/mg protein in control cells after a 100 min treatment. Finally, Na^+^ K^+^ ATPase and Ca^2+^ Mg^2+^ ATPase activities were decreased by 47.5% and 33.0% as compared to the control. These data show that bacterial energy metabolism is disrupted following PEF treatment.

The production of ATP is intimately linked to the glycolysis process. PK turns phosphoenolpyruvate and ADP into ATP and pyruvate, respectively, and is one of the primary rate-limiting enzymes in glycolysis. Pyruvate is subjected to oxidative decarboxylation by the pyruvate dehydrogenase system, resulting in the formation of acetyl-CoA. Acetyl-CoA reacts with oxaloacetic acid to form citric acid, which opens the tricarboxylic acid cycle (TCA). All aerobic organisms depend on the TCA cycle as a crucial metabolic process, which it shares with a range of anabolic pathways and the energy found in the mitochondria [40]. SDH is an enzyme in the TCA cycle, which, in both prokaryotic and eukaryotic organisms, links the respiratory electron transport chain to the TCA pathway. Additionally, it plays a crucial role in the energy metabolism of cells [41,42], and its activity is a sensitive indicator of damage to the energy metabolic systems [43]. In most microorganisms and animals, the enzyme comprises four subunits [44]. This enzyme catalyzes succinic acid to fumaric acid and simultaneously produces FADH_2_. FADH_2_ is an electron donor of the respiratory chain.

Adenosine Triphosphate (ATP) is closely related to energy metabolism, cell function, and life activities [45]. ATPase catalyzes ATP to ADP and thus supplies energy for cells [42]. The ATPase is a ubiquitous pump in the cell membrane of most cells, which is essential to establish and maintain the resting potential inside and outside the cell membrane. The establishment of an electrochemical gradient across the plasma membrane is vital for cell functions as diverse as the propagation of nerve signals, volume regulation, nutrient absorption, and pH regulation. Ca^2+^Mg^2+^ ATPase, as a calcium pump on the cell membrane, hydrolyzes ATP to pump intracellular calcium ions to extracellular to ensure the normal functions of cells. The decrease in the activity of Ca^2+^Mg^2+^ ATPase might cause the accumulation of intracellular Ca^2+^, which could then stimulate the increase in intracellular ROS levels and cytoplasmic dysfunction [46]. The function of one membrane protein, the Na+ K+ ATPase, is to maintain the permeability of the cell membrane, the low-sodium and high-potassium environment within the cell, and the resting potential [47]. Moreover, Na+ K+ ATPase would lead to the difference in H+ gradient inside and outside the cell membrane, thereby changing the permeability of the cell membrane [48].

Together, the key enzymes (PK, SDH, and ATPase) were used to evaluate the influence of PEF on respiration and energy metabolism of *E. coli*. The results showed that all of them were significantly inhibited by PEF, indicating that one of the mechanisms by which PEF inhibited *E. coli* was to disrupt the rate of intracellular metabolism and result in energy limitation.

### 3.5. Transcriptomic Analysis

#### 3.5.1. Sequence Analysis

After filtering through the raw reads, the Q20 values were >98%, and the Q30 values were >95%, suggesting that the data were of high quality, confirming successful library construction and RNA sequencing. Detailed data are shown in Table 3.

#### 3.5.2. Identification of Differentially Expressed Genes

The differentially expressed genes (DEGs) in response to the treatment with the PEF and the control from *E. coli* were identified. The criteria of DEGs were |log_2_FC| ≥ 1.0 and *p*-adjust < 0.05. Gene expression level correlation is an indicator to test the credibility of the experimental results and the reasonableness of the samples. As illustrated in Figure 5A, the correlation coefficients between the samples in the control and PEF groups were small (less than 0.697), with large differences between groups; however, the correlation coefficients for the samples within the control or PEF groups were high (both greater than 0.984), with small differences within groups. The expression patterns were similar between the parallel samples; however, the expression patterns of the groups before and after PEF treatment were more dissimilar. Furthermore, the volcano plot displayed the gene expression comparison between PEF and control samples (Figure 5). A value of log_2_FC greater than 1 value for the division of gene expression between the two groups indicates gene up-regulation, and a value of log_2_FC less than −1 value indicates gene down-regulation. There were 1537 significant DEGs (|log_2_FC| ≥ 1.0 and *p*-adjust < 0.05), including 722 up-regulated and 815 down-regulated genes.

#### 3.5.3. Gene Ontology and Kyoto Encyclopedia of Genes and Genomes Enrichment Analysis

Xiu [49] describes Gene Ontology (GO) analysis as an internationally accepted approach for describing gene function. It uses specified concepts and regulated language to expose gene features and products in every organism. Among the top 20 functional enrichment results in Figure 6A,B, the DEGs were found to be mainly in (up-regulated) alpha-amino acid biosynthetic process, amino acid biosynthetic process, carboxylic acid biosynthetic process, ATP-binding cassette (ABC) transporter complex, etc., and (down-regulated) SOS response, carbohydrate catabolic process, nucleic acid metabolic process, cellular response to external stimulus, etc. The Kyoto Encyclopedia of Genes and Genomes (KEGG) is a biological systems database that integrates genomic, chemical, and systemic functional information [7,50]. Based on bioinformatics databases, DEGs were categorized into different pathways by KEGG pathways significant enrichment analysis (Figure 6C,D). KEGG analysis revealed 20 KEGG pathways, among which, sulfur metabolism (map00920), ABC transporters (map02010), phosphotransferase system (PTS) (map02060), and galactose metabolism (map00052), and other major pathways included valine, leucine and isoleucine biosynthesis (map00290), phosphonate and phosphinate metabolism (map00440), cysteine and methionine metabolism (map00270), RNA degradation (map03018), fructose and mannose metabolism (map00051), pyrimidine metabolism (map00240), and so on.

#### 3.5.4. DEGs Related to Cell Structures

The cell membrane serves as a protective barrier for bacteria and plays a crucial role in maintaining cell integrity and shape. Proteins (60%–70%) and lipids (majoritively phospholipids, 20%–30%) make up the composition of bacterial membranes [51]. Compared with the control group, the expression of DEGs involved in proton-transporting ATP synthase complex (GO:0045259) was up-regulated after PEF treatment. The same result was found in the molecular reaction mechanism of vanillin resistance to *E. coli* studied by [52]. It is a crucial mechanism for *E. coli* to directly transform the membrane potential into usable chemical energy that this protein on the cell membrane of the bacteria may convert the energy inherent in the transmembrane electrochemical gradient into the energy of covalent phosphate anhydride bonds [53]. Treatment with PEF can damage the cell membrane, which may destroy the function of ATP synthase or disrupt the ion balance inside and outside the cell, causing the cell to produce more ATP synthase to maintain normal cell function [54]. Moreover, the 18 DEGs associated with the cell membrane for the metabolism of glycerophospholipid (map00564) have been found to have significant down-regulation. Glycerophospholipids comprise critical components of the dual-membrane envelope of Gram-negative bacteria and participate in many processes [55]. Compared with the control group, many DEGs related to pathways of cell wall structures were significantly down-regulated in response to PEF, including *mrcA* for peptidoglycan biosynthesis (map00550), *kdtA*, *lpxP*, and so on for lipopolysaccharide biosynthesis (map00540) (Figure 7A).

#### 3.5.5. DEGs Related to DNA Replication and Repair

During PEF treatment, the cell membrane was irreversibly damaged, causing leakage of vital substances such as nucleic acids and proteins [56]. In this study, the expression of DEGs related to DNA replication initiation (GO:0006270) and DNA repair complex (GO:1990391) were significantly down-regulated after being treated with PEF. Furthermore, the 26 DEGs related to DNA damage response (GO:0006974) were down-regulated, indicating damage to its DNA from environmental insults, reducing the number of *E. coli* (Figure 7B). Thus, as a result of treatment by PEF, the synthesis of DNA was restrained.

#### 3.5.6. DEGs Related to Energy Metabolism

Molecular function, biological processes, physiological behavior, growth, and even survival of organisms depend on energy metabolism. ABC transporters (ATP-binding cassette transporters), a large class of transmembrane transporters on the cell membrane, can help bacteria utilize helpful substances and discharge harmful substances. The expression of DEGs related to ABC transporters (*pstS*, *dppB*, and *aapP*) (map02010) was significantly up-regulated by 6.80 log_2_(FC), 1.07 log_2_(FC), and 1.64 log_2_(FC) after PEF (Figure 7C). The pathways for energy production in most organisms are glycolysis and the tricarboxylic acid cycle. The pathways also play a significant role in anabolism and are necessary for the synthesis of compounds such as fatty acids, carbohydrates, and aminos. Glycolysis and the tricarboxylic acid cycle (TCA) are important metabolic pathways, which are not only the central pathways of substance metabolism but also the important links of energy metabolism [40]. As shown in Figure 7D, glycolysis (map00010) and TCA cycle (map00020) in *E. coli* were significantly affected by PEF treatment. Among them, 7 of 12 glycolysis genes were down-regulated, as well as 10 of 12 TCA-related genes. It was observed that the expression levels of *D1792_24715*, *D1792_19180*, and *D1792_19175* involved in the activity of SDH (GO:0000104) were down-regulated 1.07 log_2_(FC),1.80 log_2_(FC) and 1.18 log_2_(FC) with the treatment of PEF, respectively. At the same time, it was noted that the genes (*sdhA*, *sdhC*, *sdhD*, *frdA*, *frdC*, and *frdB*) encoding the SDH for the catalysis of succinate to fumarate, the gene (*E4.2.1.2A*) for the catalysis of fumarate to malic acid and the gene (*mqo*) for the catalysis of malic acid to oxalacetic acid were significantly down-regulated in TCA cycle pathway. The DEGs (*FBP*, *ENO*, *PK*, and so on) involved in glycolysis were down-regulated. Taken together, we concluded that PEF could disrupt energy metabolism to inactivate *E. coli*.

#### 3.5.7. DEGs Related to Mobility

Bacterial chemotaxis is the response of motile bacteria to concentration gradients of different chemicals. Bacteria can prefer favorable stimuli and avoid unfavorable ones. The movement of bacterial flagella not only facilitates the initial attachment of the bacterium to the surface of an object and contributes to the preservation of the three-dimensional spatial structure of the biofilm but also facilitates the separation of the bacterium from the mature biofilm and the re-adhesion process [57]. The treatment of PEF resulted in alterations in mobility-related genes, such as the bacterial chemotaxis (map02030) genes (*mglB* and *rbsB*) and flagellar assembly (map02040) genes, being down-regulated (Figure 7E). It suggested that PEF alters the motility of *E. coli*, making it unable to avoid noxious stimuli, which may be a reason for inactivating *E. coli*.

#### 3.5.8. Quantitative Reverse Transcription PCR Verification of The Transcriptomics Sequence Results

The DEG expression levels of the same RNA samples were detected by RT-qPCR to verify the transcriptomics sequence accuracy. *16SrRNA* was selected as the reference gene. All primers used in this study are provided in Table 1. The relative expression of the gene of the PEF treatment group was compared with the control group. The value (PEF group) of the target gene was larger than that of the control gene, indicating that the target gene was up-regulated. As we can observe in Figure 6E, the expression levels of *D1792_18820* (carbohydrate metabolic process), *D1792_24705* (Succinate Dehydrogenase activity), *D1792_16885* (energy metabolism), *D1792_19100* (amino acid metabolism), *D1792_09930* (DNA damage response), *E4.2.1.2A* (Tricarboxylic acid cycle), and *rbsB* (motility) were down-regulated, while the expression levels of *D1792_19850* (membrane protein complex) and *D1792_02420* (amino acid biosynthesis) were up-regulated. Therefore, the results of the RT-qPCR assay were consistent with the transcriptome sequence results, illustrating that the transcriptome sequence was valid.

## 4. Conclusions

In conclusion, a non-thermal sterilization technology was used to evaluate the bactericidal behavior of *E. coli*. PEF was found to have excellent potential to inactivate *E. coli*. The stepwise inactivation mechanism of PEF has been made clear: PEF interacts with membranes to affect the cell morphology, damage their membrane integrity, leading to the leakage of intracellular substances, and affects the normal metabolism in the following ways: damaging DNA, protein, and enzyme inactivation (PK, SDH, and ATPase), ultimately causing the microbial cells to collapse and die. The transcriptome analysis further proved the above conclusion at the gene level. DEGs mostly related to cell structures, DNA replication and repair, glycolysis and TCA cycle, and mobility were significantly affected for PEF-treated.

*E. coli*, clearly illustrating the interaction of PEF with *E. coli*. These findings offer new perspectives on PEF’s antimicrobial mechanism and point to its possible application as a sterilization technology in the food industry to control *E. coli*. More advanced experimental technology will continue to be employed to enhance the antimicrobial activity and investigate the mechanism by which PEF is inactivated in the future.

## Figures and Tables

**Figure 1 microorganisms-12-01380-f001:**
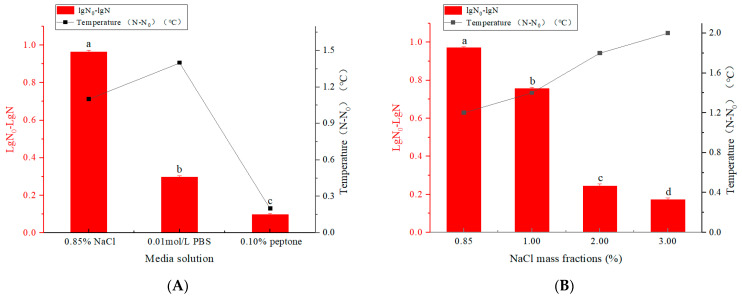
Inactivation of *E. coli* by PEF treatment. (**A**) Inactivation of *E. coli* in different media solutions by PEF. (**B**) Inactivation of *E. coli* in different NaCl mass fractions by PEF. (**C**) Inactivation of *E. coli* in different electric field intensities by PEF. (**D**) Inactivation of *E. coli* in different treatment times by PEF. (**E**) Inactivation of *E. coli* in different bacterial densities by PEF. (**F**) Inactivation of *E. coli* by PEF treatment. Different letters in the graph indicate significant differences (*p* < 0.05). Data are shown as the mean ± SD (*n* = 3), with error bars representing standard errors. Different letters in the graphs indicate significant differences (*p* < 0.05).

**Figure 2 microorganisms-12-01380-f002:**
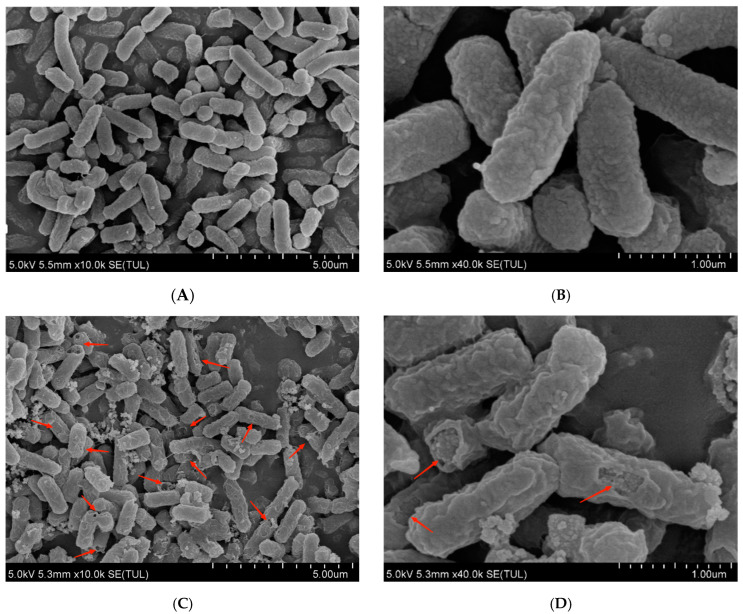
Effect of PEF on the morphology of *E. coli* (FESEM). (**A**,**B**) Control. (**C**,**D**) PEF. The red arrows indicated the damaging bacteria.

**Figure 3 microorganisms-12-01380-f003:**
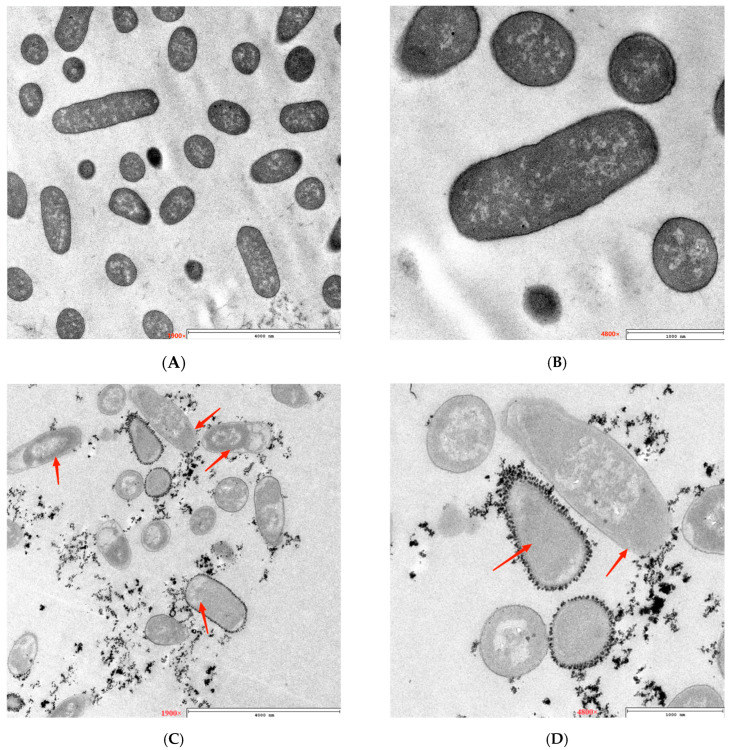
Effect of PEF on the morphology of *E. coli* (TEM). (**A**,**B**) Control. (**C**,**D**) PEF. Red arrows represent cell membrane, cell wall, and cytoplasmic changes in the cell.

**Figure 4 microorganisms-12-01380-f004:**
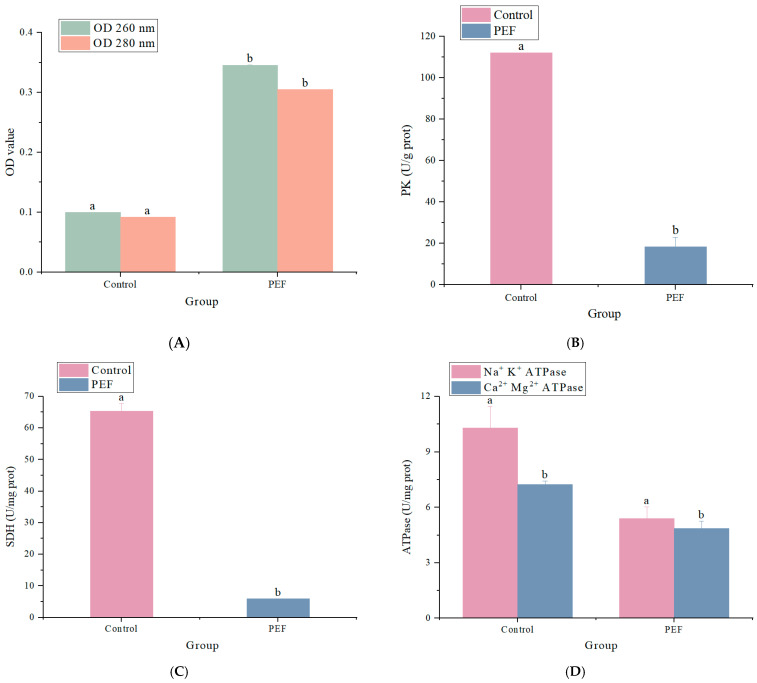
Effect of PEF on ATPase, SDH, and PK. (**A**) Effect of PEF on the cell membrane permeability of *E. coli*. (**B**) The effect of PEF on PK in *E. coli*. (**C**) The effect of PEF on SDH in *E. coli*. (**D**) The effect of PEF on Na^+^ K^+^ ATPase, Ca^2+^ Mg^2+^ ATPase in *E. coli*. Data are shown as the mean ± SD (*n* = 3), with error bars representing standard errors. Different letters in the graphs indicate significant differences (*p* < 0.05).

**Figure 5 microorganisms-12-01380-f005:**
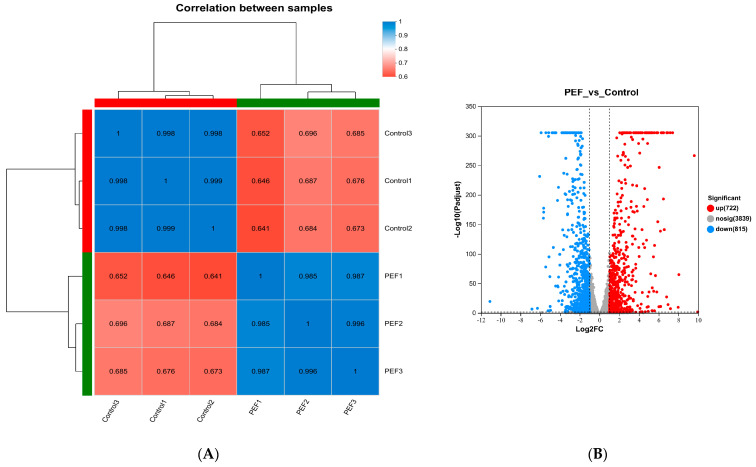
Gene expression analysis. (**A**) Correlation coefficient heat map. (**B**) Volcano plot of DEGs between PEF and control groups. The green and red colours of the horizontal and vertical coordinates in (**A**) indicate clustering.

**Figure 6 microorganisms-12-01380-f006:**
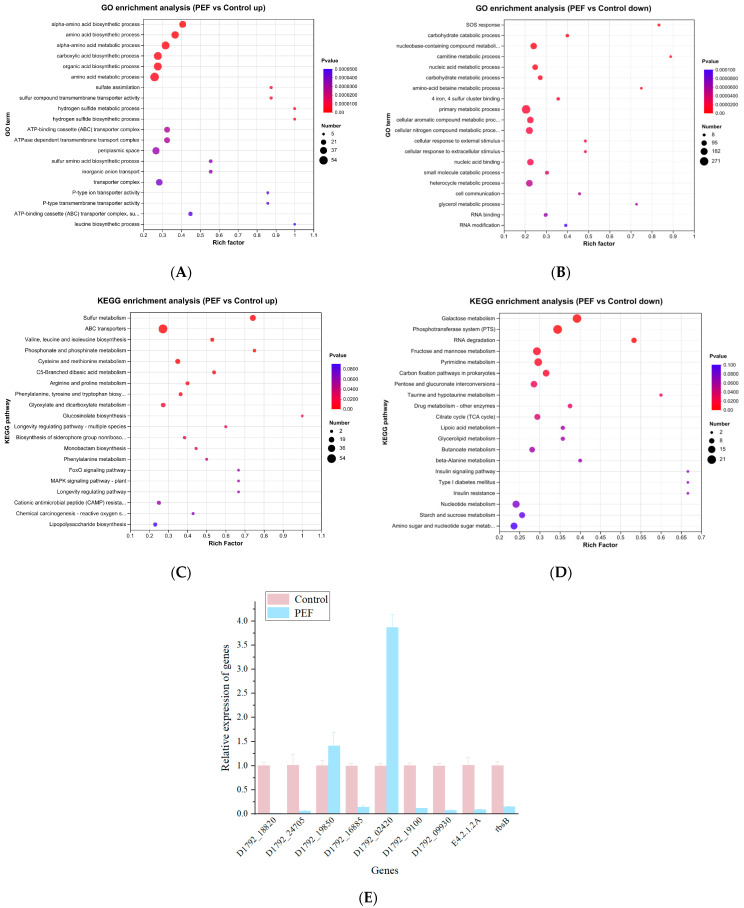
GO and KEGG enrichment analysis. (**A**) GO enrichment analysis (up-regulated). (**B**) GO enrichment analysis (down-regulated). (**C**) KEGG enrichment analysis (up-regulated). (**D**) KEGG enrichment analysis (down-regulated). (**E**) RT-qPCR verification of gene expression levels.

**Figure 7 microorganisms-12-01380-f007:**
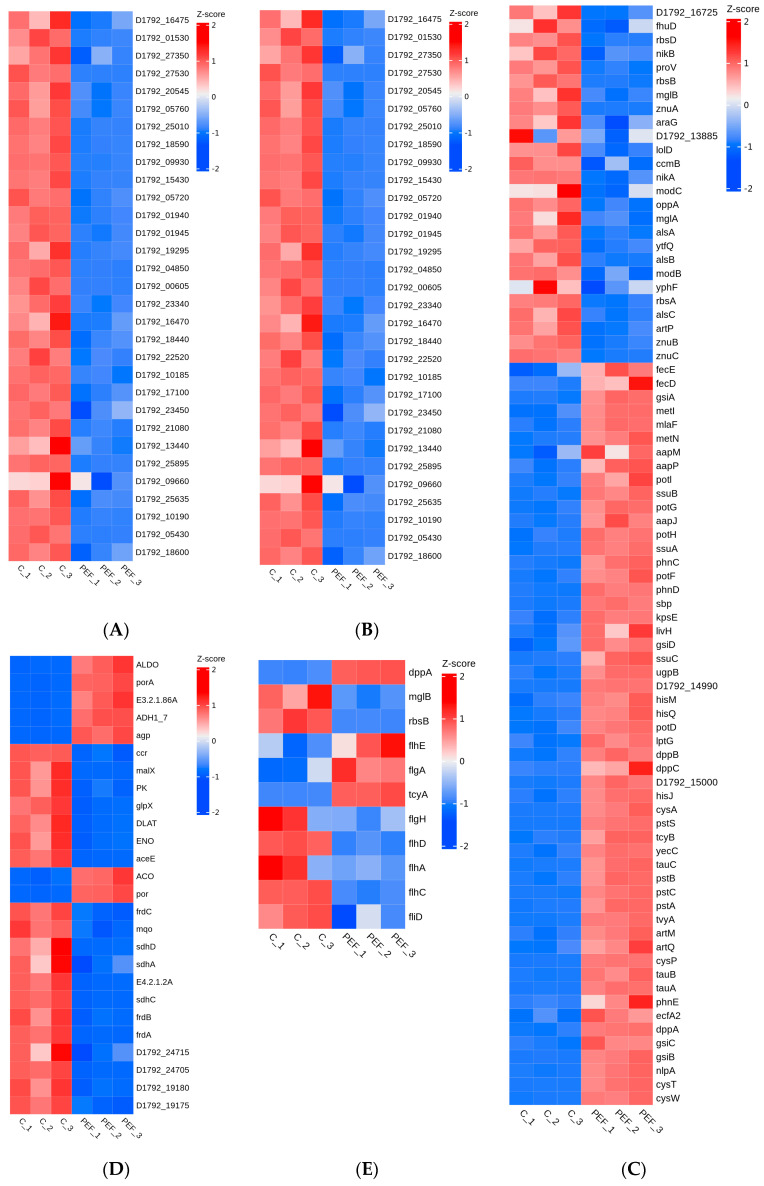
Heat map of DEGs associated with GO and KEGG enrichment analysis in *E. coli*. (**A**) Cell structures. (**B**) DNA replication and repair. (**C**) ABC transporters. (**D**) Glycolysis and TCA cycle. (**E**) Mobility. Red indicates up-regulated genes, while blue indicates down-regulated genes.

**Table 1 microorganisms-12-01380-t001:** Primer information.

Genes	Forward (F)	Reverse (R)
*16SrRNA*	TGACGTTACCCGCAGAAGAA	ATCTCTACGCATTTCACCGCTAC
*D1792_18820*	AGCCGCATCGTGGAGTGG	TCGCTTCAGATACCGCCAGAG
*D1792_24705*	CGTAGGTATTCGCCACATGATGATG	GAAAGCACGACAGTAATAACAAAGGAG
*D1792_19850*	ACTGCGATCAACTGGTGGTAATG	CCGCTTCCACGCTGAATACTG
*D1792_16885*	GGTGATGGCGTACTGGAGATATTG	AGGTTGAAACGGCGGATTTGG
*D1792_02420*	CGGCATCAATACTTACGCTCAGG	ACATACTTTATTCTCACCCAGCAACAC
*D1792_19100*	TGGCAGCAGAAGCAGGTCAG	AGCAAGCGTTGGTCAGAATGTG
*D1792_09930*	TTTGATGAAGTGGATGTAGGGATTAGC	ACCTGAGTTGATTCGCCAAGTTG
*E4.2.1.2A*	CTCCTGTTCTGCTGACCGTAATATC	ACCGCTTCGCCTTCTCCTG
*rbsB*	CGTCAGTGCGAATGCGATGG	CCACCAGGTTATAGCCAAGTTTATCC

**Table 2 microorganisms-12-01380-t002:** Conductivity of different solutions.

Different Solutions	Electrical Conductivity (mS/cm)
0.01 mol/L PBS	16.29 ± 0.00
Mass fraction of 0.85% NaCl	15.06 ± 0.01
Mass fraction of 1.00% NaCl	17.29 ± 0.00
Mass fraction of 2.00% NaCl	32.82 ± 0.02
Mass fraction of 3.00% NaCl	47.21 ± 0.02
Mass fraction of 0.10% Peptone	0.12 ± 0.00

**Table 3 microorganisms-12-01380-t003:** Sequencing QC data statistics table.

Sample Name	Raw Reads	Clean Reads	Clean Q20 (%)	Clean Q30 (%)
Control 1	29,156,142	28,864,672	98.53	96.07
Control 2	26,599,110	26,278,048	98.51	96.03
Control 3	26,553,670	26,228,896	98.46	95.94
PEF 1	24,870,364	24,507,490	98.14	95.47
PEF 2	27,193,094	26,839,560	98.30	95.76
PEF 3	24,118,414	23,765,962	98.33	95.76

## Data Availability

The original contributions presented in the study are included in the article, further inquiries can be directed to the corresponding author.

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
