# Peer review of "Effects of PEF on Cell and Transcriptomic of Escherichia coli"

_microorganisms, 2024, doi:10.3390/microorganisms12071380_

Round 1
Reviewer 1 Report
Comments and Suggestions for Authors
Line 130. Define the temperature at which it was performed, since the PEF time depends on this parameter. In food processing, optimization of the PEF parameters is necessary for each specific pulsed electric field application. How will this be adjusted?
Line 202. The title of table 1 says Morphological characteristics of colonies and molecular biology identification results and a conductivity table is presented. The table content is not related to the table title.
The objective of using this inactivation technique is to maintain the food, but also to minimally affect the nutritional and sensory properties of the food product. Are there data on food properties under the presented SPF conditions, such as organoleptic properties (structure, color, product appearance) and the content of important compounds in the food, such as anthocyanins and phenolic compounds? Some data of this type could be incorporated and presented in the results section.
Author Response
Thank you very much for taking the time to review this manuscript. Please see the attachment.

Reviewer 2 Report
Comments and Suggestions for Authors
The purpose of the research, to investigate the mechanism of PEF action on bacterial cells, is worth attention and important in terms of application in different industries and food sterilization in particular. The authors performed various analyses to study influence of different factors on PEF effects manifistation. Besides interesting data obtained by the authors the research design, some approaches, data interpretation and discussion should be improved.
Line 33: Escherichia coli should be italic
Line 95 2.1. Reagents and Bacterial Strains:
Please, check the correct information about reagents. For example:
2 × ChamQ Universal SYBR qPCR Master Mix (Nanjing Vazyme Biotech Co)
Lines 111-112: please, provide correct information about the equipment:
MyCyclerTM Thermal Cycler - Bio-Rad instead of Wuhan Beilai Biotechnology Co., Ltd;
Rotor-Gene Q (QIAGEN, Germany) instead of Roter-Gene Q (Kaijie Enterprise Management (Shanghai) Co., 112 Ltd).
Line 142: probably "might" should be "night"
Line 154: "2.3.4. Leakage of Intracellular Material" please, provide reference to this approach
Line 180 2.3.7 Quantitative Reverse Transcription PCR (RT-qPCR) Analysis
What was the endogeneous control: which gene was used to normalize data? How many sample repeats were used in one amplification run?
Primer sequences should be presented in Materials and Methods section (so, lines 532-533 "16SrRNA was selected as the reference gene. All primers used in this study are provided in Table 3." and Table 3 should be moved to this section).
Line 192:provide primer concentration instead of volume: "0.4 μL of each primer"
Table 1 does not describe "Morphological characteristics of colonies and molecular biology identification results" and does not have any connection to the text (does not mention in the text, no description)
Please, provide short explanation why NaCl solution, PBS, and peptone solution were used as media to study influence of PEF on E.coli.
And,please, justify the choice of 0.16, 0.20, and 0.24 kV/cm PEF intensities: the difference between limits is only 0,8.
Line 211-214 "E. coli presented a higher level of inactivation in mass fraction of NaCl, and the number of microorganisms decreased by 0.96 ± 0.87% lg CFU/mL. This indicates that lower conductivity favors the bactericidal efficiency of PEF". According to Table 1 low conductivity (values are almost similar) is the characteristics of 0.01 mol/L PBS (16.29) and mass fraction of 0.85% NaCl (15.06). Than why authors define descreasing only in NaCl solution, how PBS affects bacterial numbers. Why authors suggest that conductivity is the main factor.
Line 270: When analysing "Inactivation of E. coli in Different Bacterial Density by PEF" what parameters and duration of PEF was used? If there is an association (correlation) between parameter value (for example, intensities) and cell density? Could, for example higher intensity affects/kills higher densed bacterial cells?
Line 418: "Bioinformatics analysis of transcriptomics" - the title should be changed since it is only partially bioinformatic analysis
Lines 488-489:down-regulation of DEGs encoding "DNA repair complex" and "related to DNA damage response" could be associated with the absence of bacterial cells after PEF treatment not with "errors during metabolism". But according to the previous data there are survived bacterial cells after PEF treatment (decreased number but not killed). Assuming this the expression of genes encoding enzymes of DNA repair and stress response should be, on contrary, increase.
Results of RTqPCR analysis should be expanded and explained. What the differences in gene expression, define the fold, please.
Some of the obtained results should be re-considered.
Author Response

(The authors gave the same response as above.)

Reviewer 3 Report
Comments and Suggestions for Authors
The authors describe the effect of Pulsed electric field on Escherichia coli. The work is interesting due to the extensively used equipment and potential application properties. Although the PTE method is already used in the food industry.
Below are the reviewer's comments.
In the opinion of the Reviewer, we do not use abbreviations in the title.
In the opinion of the Reviewer, when using a species name for the first time, its full name should be written, not its abbreviation.
In the opinion of the Reviewer, abbreviations that have not been explained should not be used. Especially in the abstract.
What is the ultimate goal of the work?
Is it to understand the mechanisms leading to cell death?
Is it to understand the mechanisms of cell damage caused by PEF?
Does the value of PEF parameters used in the experiment kill E. coli cells or only damage them?
Why did the Authors use only one variant of PEF?
In the description of the methodology, the authors state that they used one variant of the experiment, while Figure 1 shows the use of different variants of the experiments. What was it really like?
How did the Authors calculate the % of inactivated E.coli cells?
What do the Authors mean by "mass fraction of 0.85% NaCl"? Is this equivalent to a 0.85% NaCl solution?
What was the purpose of creating Table 1? The conductivity of water depends on the concentration of salts in it. The more NaCl in the solution, the higher the conductivity. In the opinion of the Reviewer, this table is not necessary.
In Figure 1 on the Y axis please try to use the efficiency factor (lgN0-lgN /lgN0 * 100%) maybe the results will be more readable.
It has been shown that different parameters such as electric field strength, pulse frequency, and conductivity are interdependent [25]. In the Reviewer's opinion, the presented results do not support this statement.
0.97 ±0.38%, 0.76 ± 0.68%, 0.24 ±0.01, and 0.17 ± 0.93% lg CFU/mL – what units are these? In the Reviewer's opinion this is unreadable. Is 0.97 ±0.38% lg CFU/mL 0.97 lg CFU/mL where SD 0.38% of 0.97 lg CFU/Lm i.e. – 0.36 lg CFU/mL?
High SD values ​​suggest that the results obtained are random. It is possible that this is only due to statistical records. Could the Authors include the raw results, e.g. in the supplementary material?
Fig 3. What do the red arrows in Figure 3 C and D represent? They do not overlap and in the Reviewer's opinion appear to be random.
In the Reviewer's opinion, the correct name is the Krebs cycle, or possibly the citric acid cycle, not the TCA (tricarboxylic acid cycle).
In the Reviewer's opinion, SDH is not a key enzyme in the Krebs cycle. It is only unique in terms of its location in the mitochondrion and its participation in the electron transport chain.
Some studies suggest that PTE increases the concentration of some enzymes. Have the Authors encountered such literature data?
Discussion of the mechanism of enzyme reduction was not conducted. Could the authors provide it? In the reviewer's opinion, the Authors only described the role of enzymes in the cell and that their reduction may negatively affect on bacterial cells.
The decreased activities of this enzyme would presumably impede carbohydrate metabolism, which in turn would retard cell growth or even lead to cell death. - In the Reviewer's opinion, this conclusion is too general and does not correspond to reality.
It suggested that PEF alters the motility of E. coli, making it unable to avoid noxious stimuli,
which may be a reason for inactivating E. coli.– in the Reviewer’s opinion, this statement requires a broader discussion. Previously, the Authors suggested that the main way to destroy bacterial cells is by damaging the cells. Could the authors explain how a bacterium could “escape” a wave propagating in a liquid?
Could Figure 5 be of better quality?
Could Figure 6 A,B,C,D be more legible. Maybe it should be placed on a larger scale in the supplementary materials?
In the reviewer's opinion, the presented conclusions suggest that bacterial cell destruction occurs in stages. First, destruction of the cell wall and membrane, then destruction of enzymes. Couldn't these two processes occur in parallel?
The scientific literature database for 2020-2024 contains over 2,500 articles containing "Pulsed electric field Escherichia coli". Could the Authors use newer literature?
Comments on the Quality of English LanguageMinor editorial, grammatical and stylistic errors.
Round 2
Reviewer 2 Report
Comments and Suggestions for Authors
Line 114: "BIO-RAD, America" should be Bio-Rad, USA
Line 164: (the previous version Line 154): "2.3.4. Leakage of Intracellular Material" please, provide reference to this approach.
THE AUTHORS DIDN'T CORRECT despite their answer: Response 5: Thank you for pointing this out. Corrected in red in the line 154.
Line 195: reference to Table 1 should be moved after the next sentence: The RT-qPCR primers were designed and determined by Bioengineering Co., Ltd. (Shanghai).
Information about endogenous control (16SrRNA was selected as the reference gene) and numbers of repeated reaction (each sample was run in triplicate) should be included in the section!
Lines 217-218 (the previous version lines 203-204, Table 1): The sentence “From the above Table 2, it can be seen that the size of the conductivity values of different media is different.” does not describe or explain/discuss results. THE AUTHORS DIDN'T CORRECT
THE AUTHORS DIDN'T CORRECT Please, provide information about the results of gene expression analysis. The previous comment:
What the differences in gene expression, define the fold, please. In what times (2, 3, 10 etc) gene expression differ (up- or down-regulated). Define at least some differences, not just increasing/decreasing of gene expression.
The importance of presented research is undoubtful but presentation and explanation of obtained data and results should be improved.
Author Response
|
Thank you very much for taking the time to review this manuscript. Please find the detailed responses below and the corresponding corrections highlighted changes in the re-submitted files Comments 1: Line 114: "BIO-RAD, America" should be Bio-Rad, USA |
||||||||||||||||||||||||
|
Response 1: Thank you for pointing this out. We agree with this comment. Corrected in red in the line 33.
|
||||||||||||||||||||||||
|
Comments 2: Line 164: (the previous version Line 154): "2.3.4. Leakage of Intracellular Material" please, provide reference to this approach. |
||||||||||||||||||||||||
|
Response 2: Thank you for pointing this out. If the bacteria membrane is compromised,release of cytoplasmic constituents of the cell can be monitored.Nucleic acids contain pyrimidine and purine rings, etc., with characteristic peaks at 260 nm, and proteins contain tryptophan and tyrosine with characteristic peaks at 280 nm, and the concentration increases with absorbance values .Through the detection of absorbance at 260 nm and 280 nm, one can estimate the amount of DNA and proteins released from the cytoplasm. This reference is in 2.3.4.
|
Reviewer 3 Report
Comments and Suggestions for Authors
In the reviewer's opinion - Effects of pulsed electric field (PEF) on cell and transcriptomic of Escherichia coli
In the reviewer's opinion, only Escherichia coli is sufficient in the abstract in line 12, the others may be E. coli.
In the Reviewer's opinion, the purpose of the work must be clearly indicated in the manuscript, not in the response to the Reviewer.
Lack of methodology. All variants of the experiment used must be described in the material and methods section, not in the discussion of the results.
In the Reviewer's opinion, Table 2 is not necessary. It can be described in one sentence, you don't need to create a table for it.
In the reviewer's opinion, the term "mass fraction of 0.85% NaCl" is therefore a mass percentage
The results were expressed as lgN0-lgN, where N0 and N denoted the number of microorganisms (CFU/mL) before and after PEF treatment, respectively. - The Reviewer understands the expression lgN0-lgN. Comment No. 10 was about the question of whether the authors had checked the second option. In the Reviewer's opinion, the second option could be more readable for readers.
Regarding comment 16. Quote from a random article from the database - When the electric field strength is 15 kV/cm, the activity of α-amylase is the highest, and the enzyme activity is significantly increased by 17.55 ± 0.46% compared to the control. Of course, this does not have to apply to enzymes and the conditions of experiments carried out by the authors. The question was aimed at potentially expanding the scientific value of the manuscript
Regarding comment 17. What actions did the authors take?
Regarding comment 19. In the reviewer's opinion, the authors' explanation is unconvincing. PTE was applied for 100 minutes per 50 mL sample. What is the speed of movement of E. coli. What is the speed of propagation of a wave in a liquid? How often was the electrical impulse generated? How many of these impulses were there during the entire experiment?
Author Response
|
Comments 1: In the reviewer's opinion, only Escherichia coli is sufficient in the abstract in line 12, the others may be E. coli. |
|||||||||||||||||||
|
Response 1: Thank you for pointing this out. Therefore, we corrected in red in the abstract.
|
|||||||||||||||||||
|
Comments 2: In the Reviewer's opinion, the purpose of the work must be clearly indicated in the manuscript, not in the response to the Reviewer. |
|||||||||||||||||||
|
Response 2: Thank you for pointing this out. We have modified this point in red in the introduction.
|